# Iron Content of Wheat and Rice in Australia: A Scoping Review

**DOI:** 10.3390/foods13040547

**Published:** 2024-02-10

**Authors:** Yee Lui Cheung, Belinda Zheng, Yumna Rehman, Zi Yin Joanne Zheng, Anna Rangan

**Affiliations:** 1Discipline of Nutrition and Dietetics, School of Nursing, Faculty of Medicine and Health, The University of Sydney, Camperdown, NSW 2006, Australia; yche5901@uni.sydney.edu.au (Y.L.C.); bzhe0098@uni.sydney.edu.au (B.Z.);; 2Charles Perkins Centre, The University of Sydney, Sydney, NSW 2006, Australia

**Keywords:** wheat, rice, iron content, nutrient composition, Australia

## Abstract

With a shift towards plant-based diets for human and planetary health, monitoring the mineral content of staple crops is important to ensure population nutrient requirements can be met. This review aimed to explore changes in the iron content of unprocessed wheat and rice in Australia over time. A comprehensive systematic search of four electronic databases and the gray literature was conducted. A total of 25 papers published between 1930 and 2023 that measured the iron content of unprocessed wheat and rice were included. *Triticum aestivum* was the most common wheat type studied, including 26 cultivars; iron content ranged from 40 to 50 µg/g in the 1930s and 1970s and was more variable after this time due to the introduction of modern cultivars, with most values between 25 and 45 µg/g. The iron content of rice (*Oryza sativa*) was more consistent at 10–15 µg/g between the 1980s and 2020s. Variations over the years may be attributed to environmental, biological, and methodological factors but these were not well documented across all studies, limiting the interpretation of findings. As the number of individuals following plant-based diets continues to rise, the ongoing monitoring of the mineral content in commonly consumed plant-based foods is warranted.

## 1. Introduction

Globally, there are increasing movements towards food systems and patterns in favor of promoting health and environmental sustainability [1]. This is facilitated by recommendations from key authorities and government agencies, including the EAT-Lancet Commission and the World Health Organization, encouraging a shift towards plant-based diets for human and planetary health benefits through the consumption of a diverse range of plant foods and minimal quantities of red and processed meat [2,3]. There are limited data on vegetarianism in Australia; however, market research has found that the proportion of Australians following a vegetarian diet had increased from 11.2% to 12.1% between 2014 and 2019, with the largest proportion being young adults aged 18 to 45 years [4,5].

Plant-based diets, if not appropriately balanced, can result in micronutrient deficiencies, such as iron deficiency [6]. In Australia, the most significant contributors to dietary iron are cereals (31%—this category includes iron-fortified cereals); meat, game, and poultry (17%); and cereal-based products (16%—this category may include small amounts of meat products) [7]. For those not consuming animal products, there is a greater dependence on plant-based foods to maintain nutritional adequacy [6]. Iron is present in heme or non-heme forms, with heme sources being more bioavailable but only sourced from animal products [1]. It is estimated that iron absorption is approximately 18% from an omnivorous Western diet and 10% from a vegetarian diet [8]; thus, individuals following plant-based diets need to carefully plan their diets to attain their iron requirements [3].

Iron is of particular concern for females of reproductive age, given the increased requirements to account for menstrual losses [8]. In Australia, 40% of girls aged between 14 and 18 and nearly 38% of women aged between 19 and 50 have inadequate iron intakes [9]. This can be attributed to their higher requirements and lower meat consumption compared to men [9]. Furthermore, women in Western societies are twice as likely as men to be vegetarian or vegan [10]. These factors place women of reproductive age and people following plant-based diets as high-risk subgroups for iron deficiency and anemia.

Regular monitoring of the nutrient composition of everyday foods is valuable to determine any nutritional changes to the food supply. Concerns about declining mineral density in crops have been raised in overseas research examining historical changes in the nutrient composition of plant-based foods [11,12]. Of particular significance was a 2008 UK study analyzing archived wheat samples that reported decreases in zinc, copper, iron, and magnesium since the 1960s, when higher-yield semi-dwarf cultivars were introduced [11]. Further, climate change can potentially impact nutrient content, with reports that mineral concentrations in staple grains were lower under elevated carbon dioxide conditions [13,14]. The ongoing rise of plant-based diets threatens to increase the prevalence of anemia and the associated costs to populations and health systems.

Previous work in Australia investigating iron content changes in vegetables and legumes found that iron content has remained relatively consistent or slightly decreased in most vegetables [15]. Limited research has been undertaken on grains, yet flour, bread, pasta, and rice are major foods consumed by almost all Australians [7] and are a good source of energy, dietary fiber, minerals, and folate. Wheat is an important crop for both domestic consumption and export and breeding programs have been ongoing since the late 1800s to establish high-quality varieties compatible with the Australian environment [16]. In particular, the release of semi-dwarf cultivars in the mid-1970s led to significant improvements in yield [17]. Given the lack of nutritional composition of grains in Australia, this study aims to investigate changes in the iron content of wheat and rice over time. The research question is addressed with a scoping review to, firstly, identify studies examining the iron content in wheat and rice in Australia and, secondly, to assess for temporal iron content changes in these foods. The findings will provide insight into the iron levels in Australian wheat and rice and can inform future work targeted at maintaining the nutritional quality of plant-based foods.

## 2. Materials and Methods

The review was guided by the Joanna Briggs Institute Methodology for Scoping Reviews [18] and was reported in accordance with the Preferred Reporting Items for Systematic reviews and Meta-Analyses extension for Scoping Reviews (PRISMA-ScR) checklist [19], provided in Appendix A.

### 2.1. Eligibility Criteria

The inclusion criteria required articles to report numerical data on the iron content in the edible components of unprocessed wheat and rice sourced from Australia. Data were included for crops grown in standard field conditions under standard agricultural management practices and/or in baseline control groups for field experiments. The studies were limited to those written in English, but no publication date limit was set due to the nature of the research question. All articles were required to be original research studies.

Articles were excluded if they did not report numerical values of iron content; if grains were sourced outside of Australia; if grains had undergone processing involving the removal of any edible components; if inedible components of the grain were analyzed; or if grains were grown in controlled experimental conditions such as glasshouses or greenhouses. Studies investigating a treatment variable under experimental conditions were excluded if a control group was not included.

### 2.2. Information Sources and Search Strategy

A preliminary search was undertaken to identify the relevant literature. The search strategy was developed using text words and associated phrases from the relevant articles. The search strategy, including all keywords and subject headings, was adapted for each database with the aid of an academic liaison librarian (Appendix A). The following databases were systematically searched from inception to 21 August 2023: EMBASE (Ovid), CAB Abstracts via Web of Science, Web of Science Core Collections, and ANR Index and Archive (Informit). A gray literature search was conducted on 23 August 2023 through Google to identify websites of relevant and reputable organizations. Keyword string searches were undertaken to represent the research topic (wheat, iron content, Australia) and (rice, iron content, Australia), and the first 10 pages of results were screened. Handsearching of agricultural journals and Appendix A and forward and backward citation searching of key articles were conducted. All identified articles were collated and imported into EndNote 20 (Clarivate Analytics). After the removal of duplicates, articles were imported into Covidence (Veritas Health Innovation, Australia) for screening.

### 2.3. Study Selection

The articles were screened by title and abstract by two independent reviewers (Y.L.C. and B.Z.) and assessed against the eligibility criteria. Before embarking on this process, the team selected a sample of articles that were screened independently, with discrepancies discussed and modifications made to the eligibility criteria to ensure consistency between team members. Following title and abstract screening, a full-text review was undertaken of the potentially eligible articles. Any conflicts arising from the screening process were resolved by discussion with a third reviewer (A.R.).

### 2.4. Data Charting

A data-charting spreadsheet was created in Google Sheets for the data extraction of key information following the extraction framework guidelines recommended by the Joanna Briggs Institute. Two reviewers (Y.L.C. and B.Z.) independently charted data on the iron content of unprocessed wheat and rice in Australia from the included sources of evidence. Key information extracted included: author/s, publication year, type of wheat or rice analyzed, sampling parameters (study design, component of wheat or rice, collection date and location, number of samples collected), iron content (mean, standard deviation or range, converted to µg/g), and analytical method used. For sources that did not specify the date of collection or sampling, the date was assumed as the year of publication. Any discrepancies were discussed with a third reviewer (A.R.).

### 2.5. Data Synthesis

The sources were collated according to wheat or rice type and tabulated based on time points and key characteristics. The mean iron content was calculated based on reported values within each study and could combine multiple cultivars, field locations, or timing. Bubble charts were created to visualize the number of sources published by year and location in Microsoft Excel 2023 (Version 16.72). Scatterplots were produced by plotting the iron content of wheat or rice (using mean content within each study) over time (years).

## 3. Results

### 3.1. Study Selection

The search yielded a total of 5149 articles; 5145 were identified from literature databases and 4 were identified from gray literature searches (Figure 1). Following the removal of duplicates, 4916 articles were screened by title and abstract, which excluded 4820 articles. Full-text screening of 96 articles excluded a further 71 articles. Reasons for exclusion included samples grown outside Australia; not reporting numerical values of iron content; samples grown under experimental conditions; the source not reporting primary data; or the source having insufficient sampling details to determine the components of the grain analyzed. Of the 25 remaining articles, 19 analyzed wheat, 8 analyzed rice, and 2 analyzed both wheat and rice.

### 3.2. Study Characteristics

The articles sourced were published between 1930 and 2023, with most studies conducted between 1980 and 2020. Notably, no studies were published between 1932 and 1974. For wheat, most samples originated from New South Wales (NSW, n = 10), followed by Victoria (n = 9) and Western Australia (WA, n = 8), with the earliest samples analyzed in the 1930s (Figure 2). For rice, four studies examined samples from NSW, whilst the remaining studies did not specify the exact state (Figure 3). The earliest rice sample was analyzed in 1982.

Table 1 summarizes the key characteristics of studies that analyzed the iron content of wheat. Two common wheat varieties, *Triticum aestivum* (bread wheat) and *Triticum durum* (pasta wheat), including twenty-three different cultivars, were studied. The samples were obtained cross-sectionally—directly from the field; from farmers, depots, or food retailers; or from field experiments—and were analyzed as either single samples or composite representative samples. Table 2 summarizes the key characteristics of studies that analyzed the iron content of the rice *Oryza sativa*, which included six identified cultivars. Most samples were collected from food retailers or farms. For both wheat and rice samples, the exact growing location and conditions such as rainfall, temperature, and soil type were not consistently reported across studies, but standard agricultural practice at the time was assumed.

The analytical method utilized to measure iron content has changed over time, with thiocyanate colorimetry used in the 1930s; atomic absorption spectrophotometry (AAS) used in the 1970s/1980s; and inductively coupled plasma (ICP) atomic emission spectroscopy (ICP-AES), ICP-optical emission spectroscopy (ICP-OES), and ICP-mass spectrometry (ICP-MS) used from the 1990s onwards.

The reporting of iron content and its statistical variability was inconsistent between studies, with some reporting means, standard errors, standard deviation, or ranges. The mean iron content between studies ranged from 23.6 to 76.3 µg/g for wheat (23.6–76.3 µg/g for *Triticum aestivum* and 47.3 µg/g for *Triticum durum*) (Table 1) and from 8.0 to 26.0 µg/g for rice (*Oryza sativa*) (Table 2). More detailed data are available in Appendix A.

### 3.3. Temporal Changes in Iron Content of Wheat and Rice

Figure 4 illustrates the mean data points from 19 studies across 12 time points, from 1931 to 2018, reporting the iron content of the wheat variety *Triticum aestivum*. As only one study reported on the iron content of *Triticum durum*, this was excluded from the temporal analysis. Iron content was relatively consistent in the 1930s and 1970s at 40–50 µg/g. After the introduction of semi-dwarf wheat cultivars, with the exception of some high-value outliers, most data points ranged between 25 and 45 µg/g.

For rice, the mean data points of eight studies examining *Oryza sativa* over 41 years were included, with the first sample reported in 1982. Figure 5 illustrates the reported iron content over time.

## 4. Discussion

This review scoped the scientific and grey literature on the iron content of Australian wheat and rice. A total of 25 studies were identified between 1930 and 2023, but only a few studies were undertaken prior to 1980. The extracted iron data for wheat showed some consistency between the 1930s and 1970s, at 40–50 μg/g, but increased in variability after this time, with most values ranging between 25 and 45 μg/g after 1980. For rice, most studies reported iron values between 10 and 15 μg/g over the past 50 years. Several outliers with high values were reported for wheat and rice, resulting in three-fold differences in mean iron content: wheat from 24 to 76 μg/g and rice from 9 to 26 μg/g. These differences may be attributed to biological factors, environmental conditions, and sampling or analytical techniques. However, these variables were not well reported in all studies, thus limiting the interpretation of the findings.

The iron content of wheat and rice reported in this review is comparable with data reported in Australian reference food composition databases over time. Wholemeal wheat flour iron ranged from 30 µg/g in 1992 [45] to 28 µg/g in 2020 [46], whereas raw brown rice iron changed from 12 µg/g [45] to 8 µg/g [46]. Although food composition databases are constrained by various caveats, including data being derived from a composite sample subject to environmental and biological confounding factors [12], this consistency in iron content reinforces the reliability and validity of our findings.

Findings from this review can be compared to overseas studies, including those conducted in Europe and the United States. One of the world’s longest-running field studies is the Broadbalk Continuous Wheat Experiment, commenced in Rothamsted, England, in 1843 [47]. Fan et al., analyzed archived samples from this experiment and found that mean concentrations were 23–27% lower between 1968 and 2005 compared to the previous period (between 1845 and 1967), coinciding with the introduction of higher-yielding cultivars in the 1960s [11]. In the United States, Garvin et al., (2006) examined 14 common wheat varieties (hard red winter wheat) representing the production eras between 1830 and the late 1990s. The study found a negative relationship between yield and iron concentration, and that newer, higher-yielding cultivars contained lower iron levels [48]. Similarly, Murphy et al., investigated 56 historical cultivars and 7 higher-yielding modern wheat cultivars in the US and reported that modern cultivars had significantly lower iron mineral concentrations compared with older cultivars (32.3 vs. 35.7 μg/g) [49]. However, the presence of several modern high-yielding cultivars with high micronutrient concentrations indicated that this is not always the case due to genetic variability. In Australia, modern semi-dwarf wheat varieties were introduced in the mid 1970s, and subsequent adaptations and improvements in cultivars and agricultural practices over the years have led to further yield gains (Brennan). The three studies that analyzed the iron content of wheat grown prior to 1980 in Australia [20,21,22] reported mean iron values between 42 and 49 μg/g, while the majority of values were between 25 and 45 μg/g after 1980.

Relatively few studies have been conducted on the nutritional aspects of rice, despite it being a staple food across the world [50]. An analysis of brown rice samples in Asia conducted by the International Rice Research Institute found that iron concentration ranged between 6.3 and 24.4 μg/g, with the mean iron content in traditional varieties observed to be slightly lower compared to newer varieties [51]. These concentrations are consistent with those reported in the current review.

It has been suggested that selected cultivars, evolved to emphasize high yield, rapid growth, and pest resistance, may have led to limitations in their ability to extract minerals from soil, or affect transport pathways within the plant [12,52,53]. A trade-off between yield and nutrient content described as the “dilution effect” has been commonly reported to explain apparent mineral content declines [52]. The dilution effect hypothesizes that an increased accumulation of starch in the grain endosperm of higher-yielding cultivars results in decreased nutrient concentrations [38]. Shorter-term studies conducted after the 1960s have not reported changes in iron content in grains—for example, a Finnish study examining wheat flour and brown rice obtained in 1980 and 2007 found no difference in iron content [54]. Our review identified 23 different wheat cultivars for *Triticum aestivum* and 6 rice cultivars for *Oryza sativa*, but not all studies reported the specific cultivars that were sampled.

Although genetic variation is thought to have the greatest impact on iron concentration, environmental factors such as climate, soils, and agricultural practices also contribute [53]. The review included grain samples across different time periods and geographic locations, which were thus exposed to various growing conditions, reflective of the common cultivars farmed in that sampling period. Wheat-growing regions throughout Australia vary widely in terms of soil and climate; growing areas in Queensland and New South Wales include tropical, sub-tropical, and temperate environments with relatively high rainfall and vertosol clay soils, whereas Western Australia, with a more Mediterranean climate and low soil fertility, is dependent on winter rainfall [55]. The soil profile, characterized by micronutrient levels, soil type, and pH, can influence the uptake and nutrient availability within the grain [56]. For example, clay soils typically contain a higher concentration of mineral particles compared to sandy soils [57], and iron availability in soil decreases with increasing pH [58]. The historical debate concerning soil depletion leading to reductions in crop mineral content has now been refuted with the widespread implementation of soil testing and fertilizers under conventional agricultural practice [12]. However, exposure to environmental stressors such as waterlogging, herbicide damage, and excessive fertilizer use can affect grain mineral content [22,56,59]. Temperature and carbon dioxide levels are other factors that impact the mineral composition of wheat and rice, with studies having linked elevated carbon dioxide and temperatures to reductions in iron content [14,60,61,62]. The impact of climate change on the nutritional composition of foods is currently an active area of research [62].

This review revealed a significant degree of heterogeneity in sampling methods across studies. The analytical method of measuring iron content has evolved over time. Early wet chemistry methods with low sensitivity, such as thiocyanate colorimetry, were replaced by chemical analytical instruments such as AAS in the 1970s [12]. This was followed by a transition to ICP methods, including ICP-AES, ICP-OES, and ICP-MS, after the 1990s, with improved detection limits and higher precision [63]. Changes in analytical methods have resulted in improved specificity and reductions in sample contamination stemming from equipment, reagents, and soil remnants [52,64]. This presents the potential of earlier studies yielding higher values and reflecting a modest level of analytical bias.

Although our review found no evidence of a significant decline in the iron content of wheat and rice, an increased reliance on grains as a result of reduced meat consumption can raise implications for population subgroups with high iron requirements. For example, the prevalence of iron deficiency in adult women aged 25–50 years is estimated to be 20% [65], but more recent and representative data on iron deficiency are required given the popularity of plant-based diets. Apart from a person’s iron status and heme iron intake, iron bioavailability is dependent on nutritional components that act as enhancers or inhibitors of iron absorption [66]. Vitamin C consumed in the form of ascorbic, citric, or malic acid substantially increases iron absorption in the body and can also counteract the effect of inhibiting factors [66]. Conversely, nutrients such as calcium and zinc, and components such as polyphenols (in tea and wine) and phytates (in wholegrains), can inhibit iron absorption [8]. With these considerations, individuals following plant-based diets and women of reproductive age should be informed on the current evidence-based recommendations and implement strategies to maximize iron absorption. This includes consuming a vitamin C source (fruits and vegetables) with non-heme iron sources (such as grains), avoiding the consumption of foods containing inhibitory components (dairy, wine, tea, coffee) in the same meal as iron sources, and incorporating a diverse range of foods containing iron and iron-fortified products within a balanced diet [8].

Food fortification is an effective approach to combating global micronutrient deficiencies, with cereal-based foods commonly being utilized as vehicles for iron fortification [66]. Post-harvest iron-fortification programs are predominately implemented in low-middle-income countries with a higher prevalence of anemia [66] and are not currently mandated in Australia [67], although some food manufacturers are implementing the voluntary fortification of breads and cereals. The biofortification of staple crops is another approach to increasing iron in the food supply. Biofortification incorporates strategies such as breeding for micronutrient density, biotechnology and genetic engineering techniques, and agricultural practices [68]. Overseas research involving the application of grain biofortification has demonstrated some success in alleviating deficiencies in populations that are unable to attain diverse food-consumption patterns [68]. This study was strengthened by the broad range of peer-reviewed and gray literature sources, captured through a comprehensive search of electronic databases and the Google search engine with no time-frame limitations. Another strength was the use of two independent reviewers, which allowed for increased thoroughness in the study selection and data extraction process. The full texts of all included sources were able to be obtained with the aid of a liaison librarian. Limitations included a lack of sampling details, such as specific cultivars, soil type, and climatic conditions, reported in the studies. The lack of such information might increase the ambiguity of the results and interpretation as these factors could significantly affect iron concentration. However, it was beyond the scope of this review to control for environmental, biological, and methodological factors; thus, definitive reasons for iron content variations were unable to be established. In addition, data points were not representative of all Australian wheat available at the time. Another limitation was the small number of studies analyzing the nutritional composition of the individual wheat cultivars, *Triticum durum*, and rice. For example, no studies analyzing rice were published prior to 1980. This presents a gap in the literature, and further research should be directed toward investigating the iron content of other common wheat varieties and rice.

## 5. Conclusions

This scoping review presents a comprehensive exploration and assessment of iron content changes in Australian-grown wheat and rice over the years, based on the available literature. Due to the paucity of studies, variability of data points, and limitations in the reporting of biological, environmental, sampling, and/or analytical factors, no definitive conclusion on the temporal patterns of the iron content of wheat and rice is apparent. In the context of a growing population of individuals following plant-based diets, among evolving environmental conditions, technological advances, and agricultural practices, the continuous monitoring of nutrient levels in staple cereal and grain foods is recommended.

## Figures and Tables

**Figure 1 foods-13-00547-f001:**
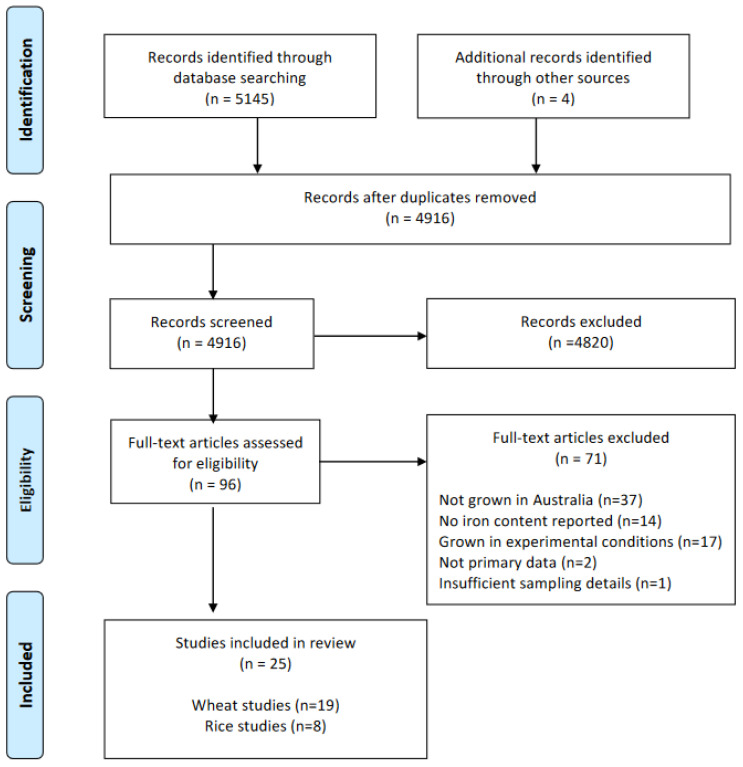
PRISMA flowchart illustrating the systematic screening process performed for the scoping review.

**Figure 2 foods-13-00547-f002:**
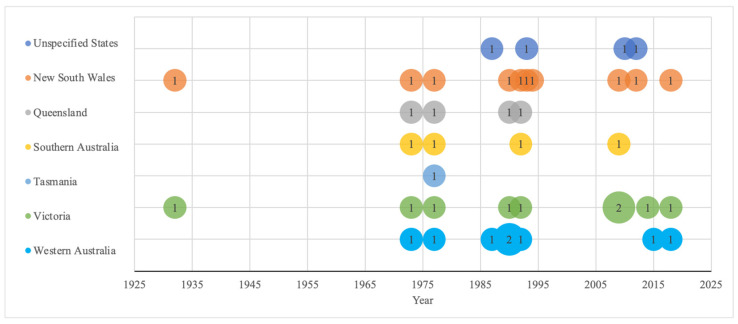
Sampling locations of wheat iron content by year from 1931 to 2018. “Unspecified States” indicates location details were not specified. Numbers within circles indicate number of studies conducted at each timepoint per state. The number of sampling locations is greater than the number of studies due to some studies sampling from more than one state.

**Figure 3 foods-13-00547-f003:**
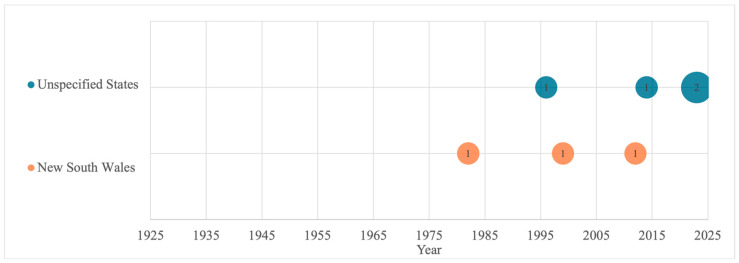
Sampling locations of rice iron content by year from 1982 to 2023. “Unspecified States” indicates location details were not specified. Numbers within circles indicate number of studies conducted at each timepoint per state.

**Figure 4 foods-13-00547-f004:**
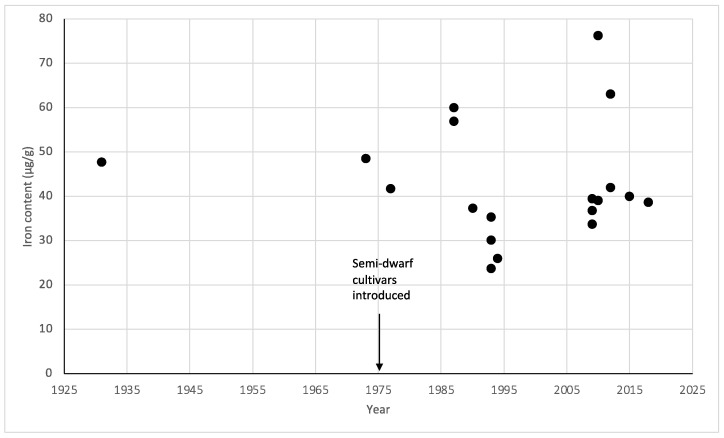
Temporal changes in the iron content of Australian-grown wheat of the variety *Triticum aestivum* as reported between 1931 and 2018.

**Figure 5 foods-13-00547-f005:**
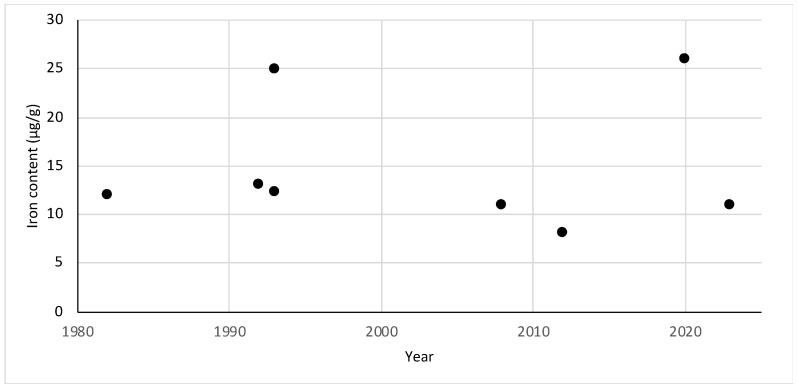
Temporal changes in the iron content of Australian-grown rice of the variety *Oryza sativa* as reported between 1982 and 2023.

**Table 1 foods-13-00547-t001:** Studies reporting iron content in unprocessed, edible components of wheat grown in or sourced from Australia.

Reference	Variety/Cultivar	Sampling Details (Year, Location, and Samples Collected)	Analysis Method	Iron Content µg/g ^a^
Dadswell, 1935 [20]	*Triticum aestivum cv*. Free Gallipoli, Nizam, Nabawa, Major, Minister, Federation, Ranee, Comeback	Collected 1930–1931, VIC and NSW n = 25	Thiocyanate colorimetry	47.8 (SD = 8.8, R = 32.0–62.0)
Murphy and Law, 1974 [21]	*Triticum aestivum*	Collected 1971–1973, NSW, QLD, VIC, SA and WA QLD samples representative of wheat delivered to major depots in Southern QLD Samples from other states representative of total wheat production in each state n = 2–78 per state, total 88	AAS	48.6 (SD = 5.4, R = 43.2–54.5)
Mugford and Steele, 1980 [22]	*Triticum aestivum*	Collected 1977, NSW, VIC, WA, SA, QLD, TAS Representative samples of wheat from 62 Australian flour mills, n = 3–24 per state, total 62	AAS	41.8 (R = 34.8–50.9)
Zarcinas et al., 1987 [23]	*Triticum aestivum* L.	Collected 1987 (publication year), AUS Field sample, n = 1	ICPS	60.0
Bolland et al., 1993 [24]	*Triticum aestivum cv. Gutha*, *Jacup*, *Eradu*	Collected 1987, WA (three locations) Field experiment controls Samples grown in three replicate plots Rainfall during growing season: 203–490 mm Annual rainfall: 330–600 mm	ICP-AES	57.0 (SD = 23.6, R = 37.0–83.0)
Batten, 1994 [25]	*Triticum aestivum* *Prime Hard*, *Australian Hard* *Australian Standard White*, *Soft* *Triticum durum*	Collected 1987–1990, NSW, QLD, SA, VIC and WA Composite samples representative of wheat grades in various port zones, n = 1–20 samples per cultivar, total 47	ICP-AES	*T aestivum*: 37.4 (SD = 2.0, R = 34.7–39.2) *T durum*: 47.3
Hocking, 1994 [26]	*Triticum aestivum* L. *cv. Egret*	Collected prior to 1994 (publication year), NSW Field experiment controls, samples grown in four representative plots	XRF	26.0
Booth et al. 1996 [27]	*Triticum aestivum Soft*	Collected 1993 Representative samples randomly collected from different batches and geographical locations in Australia, n = 4	ICP-AES	35.4 (SEM = 0.7, R = 30.0–40.0)
Morrison, 1996 [28]	*Triticum aestivum cv. Australian Prime Hard*, *Australian Hard*, *Australian Standard White*, *Australian Soft*, *General Purpose*	Collected 1990–1993, QLD, NSW, SA, VIC and WA Composite receival site samples, n = 1–33 per cultivar, year and location, total 233	ICP-AES	30.2 (SD = 6.8, R = 16.7–52.2)
Ryan et al., 2004 [29]	*Triticum aestivum cv. Vulcan*, *Janz*, *Dollarbird*	Collected 1991–1993, NSW (two locations) Field experiment controls, samples grown in 2–15 plots Ardlethan mean rainfall: 490 mm (1991—373 mm; 1992—784 mm; 1993—601 mm) Yenda mean rainfall: 420 mm (1993—554 mm)	XRF	23.8 (SD = 1.4, R = 19.0–33.0)
Fernando et al., 2012 [30]	*Triticum aestivum* L. *cv. Yitpi*	Collected 2008–2009, VIC Field experiment controls, samples grown in four replicates of two plots Growing season 2008: average 19 °C, 127 mm rain Growing season 2009: average 20.3 °C, 213 mm rain	ICP-AES	33.8 (SD = 8.8, R = 27.5–40.0)
Norton, 2013 [31]	*Triticum aestivum* L. *cv. Yitpi* *Triticum aestivum* L. *cv. Gladius*	Collected 2008–2009, NSW, VIC and SA Samples collected at random from 70 national variety trial sites across 12 regions, n = 12	ICP–OES	36.8 (SD = 6.9)
Fernando et al., 2014 [32]	*Triticum aestivum* L. *cv. Yitpi* *Triticum aestivum* L. *cv. Janz*	Collected 2007, 2008 and 2009, VIC Field experiment controls, samples grown in four replicate plots Growing season: Mediterranean climate with several 40 °C days post anthesis, 250–300 mm rain	ICP-AES	39.5
Ishida et al., 2014 [33]	*Triticum aestivum* *Australian Standard White*, *Prime Hard*	Collected 2009–2010, AUS Cross sectional analysis, n = 88	ICP-MS	39.1 (SD = 7.0)
Broom et al., 2014 [34]	*Triticum aestivum* *Australian Prime Hard*	Collected 2012, AUS Single sample from one region in Australia, n = 1	ICP-AES or ICP–MS	42.0
Rose et al., 2015 [35]	*Triticum aestivum* L. *Wyalcatchem*	Collected 2012, NSW Field experiment controls, three replicate plots	ICP-OES	63.1
Jin et al., 2019 [36]	*Triticum aestivum* L. *Yitpi*	Collected 2010, VIC Field experiment controls, Samples grown in three different soil types, four replicate plots	ICP-OES	76.3 (SD = 9.4, R = 66.4–85)
Beasley et al., 2019 [37]	*Triticum aestivum* L.	Collected 2015, WA Field experiment controls, wild type	ICP-MS	40.0
Joukhadar et al., 2021 [38]	*Triticum aestivum*	Collected 2017 and 2018, NSW, VIC and WA Field experiment controls, Samples grown in three trials of two replicate plots	ICP-MS	38.7 (SD = 1.4, R = 37.1–39.1)

^a^ Value expressed as mean with or without measure of variability, dependent on study. Mean calculated from reported values within each study (may be multiple cultivars, locations or time points). Data rounded to the nearest one decimal point. AAS: atomic absorption spectrophotometry; ICP-AES: inductively coupled plasma atomic emission spectroscopy; ICP-MS: inductively coupled plasma-mass spectrometry; ICP-OES: inductively coupled plasma optical emission spectroscopy; ICPS: inductively coupled plasma spectrometry; R: range; SD: standard deviation; SEM: standard error of mean; XRF: X-ray fluorescence spectrometry.

**Table 2 foods-13-00547-t002:** Key characteristics of studies reporting the iron content in unprocessed, edible components of brown rice grown in or sourced from Australia.

Reference	Variety/Cultivar	Sampling Details (Year, Location, and Samples Collected)	Analysis Method	Iron Content µg/g ^a^
Wills et al., 1982 [39]	*Oryza sativa* L.	Collected 1982 (publication year), NSW 4 × 500 g retail packets combined to form a composite sample of each brand, n = 2	AAS	12.0 (SD = 1.4, R = 11.0–13.0)
Marr et al., 1995 [40]	*Oryza sativa* L. *Amaroo*	Collected 1991–1992, NSW Samples obtained from individual farmer deliveries, n = 90	ICP-AES	13.0 (R = 5.0–67.0)
Booth et al., 1996 [27]	*Oryza sativa* L.	Collected 1993 Representative samples randomly collected from different batches and geographical locations in Australia, n = 7	ICP-AES	12.3 (SEM = 0.1, R = 11.5–13.0)
Marr et al., 1999 [41]	*Oryza sativa* L. *Amaroo*, *Langi*, *YRL38*, *Pelde*, *Kyema*	Collected 1993–1994, NSW Field experiment controls n = 9 (1993), n = 8 (1994)	ICP-OES	24.9 (SD = 2.1, R = 22.0–25.0)
Wurm et al., 2012 [42]	*Oryza sativa* L.	Collected 2008, NSW Pooled commercial rice samples supplied by SunRice Pty Ltd., Leeton, Australia NSW	ICP-AES	11.0
Broom et al., 2014 [34]	*Oryza sativa* L.	Collected 2012, Sample consisting of intact grains from northern and southern regions in Australia, n = 1	ICP-AES or ICP–MS	8.0
Rahman, 2023 [43]	*Oryza sativa* L.	Collected 2023 (publication year) Rice obtained from local Australian market, n = 1	ICP-MS	11.5
Birch et al., 2023 [44]	*Oryza sativa* L.	Collected 2020 Samples purchased at a major food retailer in Australia, n = 3	ICP-MS	26.0 (SEM = 0.2)

^a^ Value expressed as mean with or without measure of variability, dependent on study. Mean calculated from reported values within each study (may be multiple cultivars, locations or time points). Data rounded to the nearest one decimal point. AAS: atomic absorption spectrophotometry; ICP-AES: inductively coupled plasma atomic emission spectroscopy; ICP-MS: inductively coupled plasma-mass spectrometry; ICP-OES: inductively coupled plasma optical emission spectroscopy; R: range; SD: standard deviation; SEM: standard error of mean.

## Data Availability

Data are contained in the article and Appendix A.

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
