# Peer review of "Iron Content of Wheat and Rice in Australia: A Scoping Review"

_foods, 2024, doi:10.3390/foods13040547_

Round 1

Reviewer 1 Report

Comments and Suggestions for Authors

The authors studied the effect of time (1930 to 2023) on the accumulation of iron content in major cereals using a meta-analysis method. 

The authors revealed that time does not influence the accumulation of iron content in wheat and rice. 

Lines254: Similarly, Murphy et al., investigated 56 historical cultivars and 7 higher-yielding modern wheat cultivars in the US and reported that modern cultivars had significantly lower iron mineral concentrations compared with older cultivars (32.3 versus 35.7 μg/g) [47].

Lines 261: An analysis of brown rice samples in Asia conducted by the International Rice Research Institute found that iron concentration ranged between 6.3–24.4 μg/g, with the mean iron content in traditional varieties observed to be slightly lower compared to newer varieties.

The main question that needs attention is a comparison of the performance of old vs new cereal cultivars.  Modern cultivars are high-yielding cultivars, which may not accumulate minerals as wild types. 

The authors only discussed time and iron accumulation, but a comparison between old and new cultivars is scanty. The author should categorize all published articles into two categories, old and new, and then start analyzing data. Iron content is not too much in the endosperm of cereals, but in bran layers so only those MS which are claiming iron content in meal but not flour may be considered. 

Major Objection: I am not satisfied with the data presented in MS and the analysis carried out for such a highly relevant topic. MS may be revised to a greater extent. 

Reviewer 2 Report

Comments and Suggestions for Authors

This review paper on iron content in wheat and brown rice is an extremely valuable contribution to the scientific community.

However, what is confusing to me is that it is stated that there is no significant difference and the numbers do not support this. It would be interesting to see the change for the same culture over the years. Also, changes in the micronutrient composition are largely related to the soil, climate and feeding, so the regression line (over the years) does not seem like the best solution.

The differences are obvious and should definitely be related to the average temperatures and/or precipitation in the year before and during the harvest.

If you put the meteorological data for the mentioned years as a separate graph - it would greatly support your conclusions.

The above represents minor changes.
